# Principles of meiotic chromosome assembly revealed in *S. cerevisiae*

Stephanie A. Schalbetter [1,5*], Geoffrey Fudenberg [2,5*], Jonathan Baxter [1], Katherine S. Pollard [2,3,4*] & Matthew J. Neale [1*]

During meiotic prophase, chromosomes organise into a series of chromatin loops emanating from a proteinaceous axis, but the mechanisms of assembly remain unclear. Here we use *Saccharomyces cerevisiae* to explore how this elaborate three-dimensional chromosome organisation is linked to genomic sequence. As cells enter meiosis, we observe that strong cohesin-dependent grid-like Hi-C interaction patterns emerge, reminiscent of mammalian interphase organisation, but with distinct regulation. Meiotic patterns agree with simulations of loop extrusion with growth limited by barriers, in which a heterogeneous population of expanding loops develop along the chromosome. Importantly, CTCF, the factor that imposes similar features in mammalian interphase, is absent in *S. cerevisiae*, suggesting alternative mechanisms of barrier formation. While grid-like interactions emerge independently of meiotic chromosome synapsis, synapsis itself generates additional compaction that matures differentially according to telomere proximity and chromosome size. Collectively, our results elucidate fundamental principles of chromosome assembly and demonstrate the essential role of cohesin within this evolutionarily conserved process.

[1] Genome Damage and Stability Centre, School of Life Sciences, University of Sussex, Brighton, UK. [2] Gladstone Institutes for Data Science and Biotechnology, San Francisco, USA. [3] Department of Epidemiology & Biostatistics, Institute for Human Genetics, Quantitative Biology Institute, and Institute for Computational Health Sciences, University of California, San Francisco, CA, USA. [4] Chan-Zuckerberg Biohub, San Francisco, CA, USA. [5] These authors contributed equally: Stephanie A. Schalbetter, Geoffrey Fudenberg. *email: S.Schalbetter@sussex.ac.uk; geoff.fudenberg@gladstone.ucsf.edu; katherine.pollard@gladstone.ucsf.edu; M.Neale@sussex.ac.uk

uring meiosis, eukaryotic chromosomes are broken, repaired and paired with their homologues followed by two rounds of segregation—a series of events accompanied by dynamic structural changes of the chromosomes (Fig. 1a, top). Most prominent is the paired arrangement of pachytene chromosomes into a dense array of chromatin loops emanating from proteinaceous axes linked by a central core, the synaptonemal complex (SC), which is highly conserved across eukaryotes[2,3]. In *S. cerevisiae*, structural components include the meiotic cohesin kleisin subunit, Rec8[4], the transverse filament, Zip1[5], the axial/lateral elements, Hop1 and Red1[6,7], and the pro-DSB factors Rec114-Mei4-Mer2 (RMM)[8,9]. Rec8 is a major component of the meiotic axis—its absence disturbs the localisation patterns of Red1 and Hop1[10,11], with no axial or central elements detected by electron microscopy (EM)[4]. In the absence of Hop1 or Zip1, unsynapsed axial elements are formed[4,5]. Much of our understanding of meiotic chromosome structure has been deduced from a combination of EM, immunofluorescence microscopy and the genome-wide patterns of protein localisation determined by ChIP. However, clarifying the link between key meiotic protein complexes, chromosome conformation and genomic sequence is of great interest.

Chromosome conformation capture (3C) techniques generate maps of pairwise contact frequencies that are snapshots of chromosome organisation. 3C methods were originally applied to assay chromosome conformation in *S. cerevisiae*, including during meiosis[12]. Now they are widely used across a range of organisms and cellular contexts to link 3D organisation directly with genomic sequence[13], revealing important roles of the Structural Maintenance of Chromosomes (SMCs) cohesin and condensin in genomic organisation[14,15], where they likely mediate chromosome compaction via the process of loop extrusion[16].

Recent studies have utilised Hi-C to investigate meiotic chromosome structure in mammals[17–19] and *S. cerevisiae*[20]. Consistent

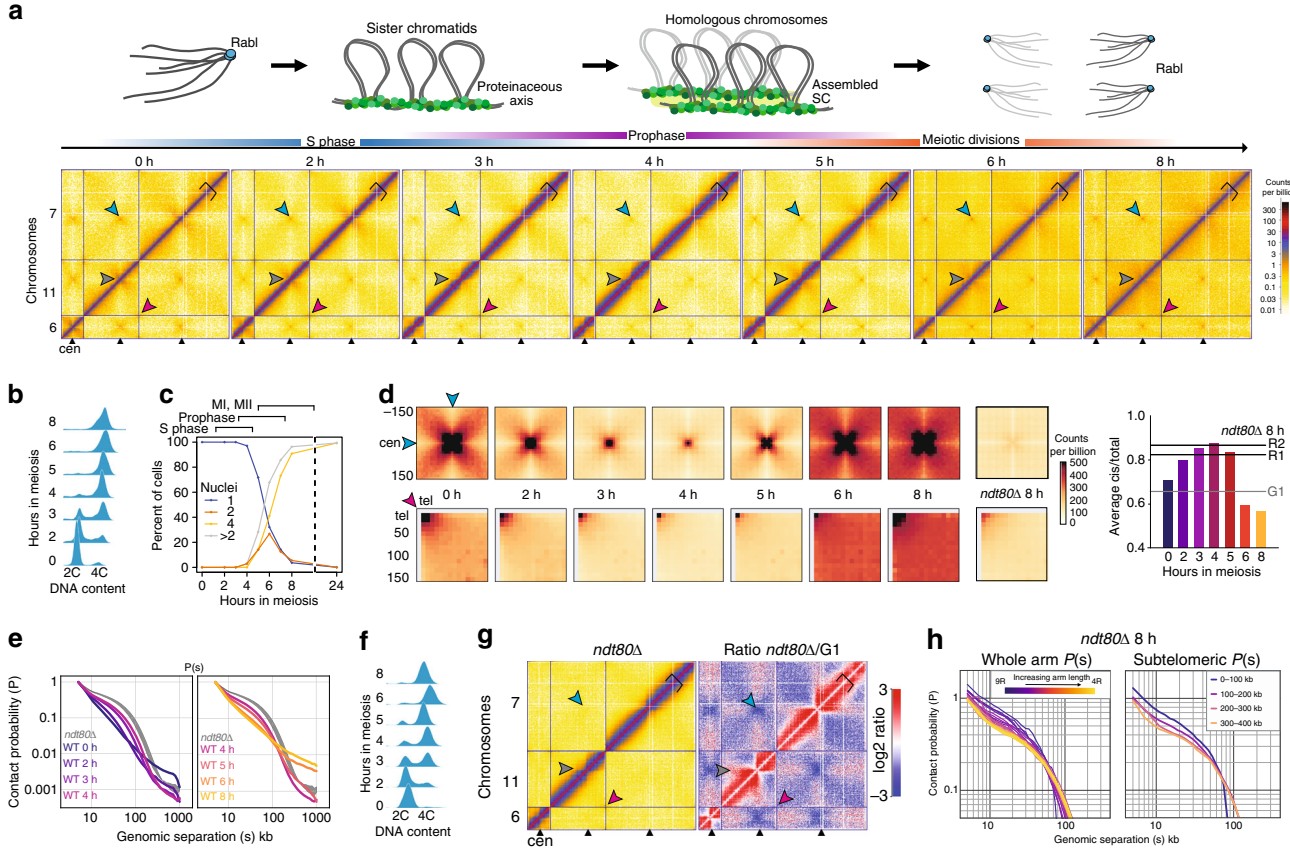

**Fig. 1** Chromosome conformation during yeast meiosis. **a** Upper panel: cartoon of chromosome morphology during the stages analysed in the meiotic time course. The Rabl-structure observed in G1 is characterised by centromere clustering (in blue), meiotic axis proteins are represented in green, with the fully assembled synaptonemal complex (SC) represented in light green. Lower panels: cells were collected during meiosis at indicated timepoints and analysed by Hi-C. At 0 h the cells are in G1. Representative Hi–C contact maps of chromosomes 6, 11 and 7 plotted at 5 kb resolution. Centromeres, telomeres and arm fold-back at the centromere are indicated by blue, red and grey arrows, respectively, and axial compaction by the width of the main diagonal relative to the fixed-width black clamp. For interactive HiGlass[1] views see: http://higlass.pollard.gladstone.org/app/?config=Z5iwKpjzQpePCXXyvuYGeQ. **b** Meiotic entry assessed by FACS; at 4 h, the majority of cells show a 4C peak indicating completion of DNA replication. **c** Meiotic progression was monitored by quantification of nuclear divisions determined by DAPI staining. Around 4 h, cells start to undergo meiotic divisions I and II. The majority of cells undergo meiotic divisions between 4 h and 8 h, indicating the degree of heterogeneity within the cell population. Source data are provided as a Source Data file. **d** Upper panels: Average *trans* centromere–centromere contact maps. Lower panels: *trans* telomere–telomere contact maps. Right: ratio of *cis* to total contact frequency. **e** Intra-arm contact probability versus genomic distance, *P(s)*, indicating the emergence (left) and disappearance (right) of chromosome arm compaction during meiosis. Shaded area bounded above and below by the two *ndt80Δ* 8 h replicates. **f** Meiosis was induced in *ndt80Δ* cells for 8 h and meiotic entry was checked by monitoring DNA replication by FACS. **g** *ndt80Δ* cells were grown for 8 h in sporulation media and analysed by Hi-C (left). Log2 ratio of *ndt80Δ* cells 8 h over G1 (right). Centromeres and telomeres are indicated by blue and red arrows, respectively, and axial compaction by a black clamp. **h** Left: contact probability of individual chromosome arms stratified by length. Right: contact probability stratified by the distance from the telomere

with compaction by a loop array, pachytene chromosome structure displays increased short-range cis interactions and a shoulder in contact frequency versus distance curves[17–20]. Mouse and monkey pachytene chromosomes additionally display a loss of the topological domains (TADs) characteristic of mammalian interphase[17–19]. In *S. cerevisiae*, Hi–C with a synthetically re-designed chromosome found low levels of interhomologue contacts, and increased insulation at Rec8 sites[20]. However, it remains to be determined whether cohesin is required for the formation of meiotic chromosome structure, as measured by Hi-C, and what mechanisms organise meiotic chromosomes.

Here we employ yeast meiosis as a model system to elucidate mechanisms of chromosome assembly, and define the role of key meiotic chromosome components, including cohesin and the SC. We show that meiotic chromosome compaction is accompanied by the emergence of punctate grid-like interactions. These interactions are dependent on Rec8 and their underlying DNA loci are preferred Rec8 association sites. Our data agrees with polymer simulations of loop extrusion with barriers, which suggest a remarkable heterogeneity in loop size and location from cell-to-cell. We further show that the synaptonemal complex modulates compaction differentially along chromosome arms.

## Results

**Chromosome compaction emerges and subsides in meiosis.** Starting with a synchronised G1 population we analysed timepoints encompassing DNA replication, meiotic prophase and both meiotic divisions (Fig. 1a–c, Supplementary Fig. 1a–c). In G1, we detect strong centromere clustering (Fig. 1a, d) and folding back of the arms at the centromeres (Fig. 1a, Supplementary Fig. 2), characteristic of a Rabl conformation[12,21]. During meiosis, centromere clustering is transiently dissolved (3–5 h, Fig. 1a, d, Supplementary Fig. 1a); this coincides with a global decrease in inter-chromosomal contact frequency at mid-prophase, reflecting chromosome individualisation. Subtelomeric clustering also decreases during meiotic prophase (Fig. 1a, d, Supplementary Fig. 1a, Supplementary Fig. 3). Our wild-type timecourse displayed no evidence of a telomere bouquet, likely due to its transience, which has been measured by microscopy[22].

Entering meiosis, contact frequency versus distance, *P(s)*, curves display a shoulder, consistent with the linear compaction of chromosome arms increasing due to *cis*-loop formation (2–4 h, Fig. 1e, Supplementary Fig. 1d, e.g. as defined[23]; for review[16]). This change in *P(s)* is reminiscent of the SMC-dependent changes observed via Hi-C during mitosis across species[24–28]. Compaction coincides with meiotic prophase I and the formation of the SC at pachytene, and is lost at later stages (Fig. 1e, Supplementary Fig. 1d).

To study meiotic chromosome conformation in more detail, and to eliminate cell-to-cell heterogeneity (Fig. 1b, c), we enriched for pachytene cells in subsequent experiments by inactivating Ndt80, a transcription factor required for exit from meiotic prophase[29]. *ndt80Δ* cells entered meiosis synchronously, assessed by bulk DNA replication (Fig. 1f), but do not initiate the first nuclear division[29]. Similar to the wild-type prophase population (3–5 h), but likely accentuated by the increased homogeneity, Hi-C maps of pachytene-enriched cells (Fig. 1g) displayed total loss of centromere clustering (Supplementary Fig. 2) and dramatic chromosome arm compaction (Fig. 1e). Shorter chromosomes (Supplementary Fig. 1e) and shorter chromosome arms (Fig. 1h, Supplementary Fig. 1f), displayed elevated contact frequency at short genomic separations, and an earlier shoulder. These features may arise from the distinct behaviour of subtelomeric and subcentromeric regions (Fig. 1h, Supplementary Fig. 1g). Alternatively, or in addition, distinct *P(s)* for chromosomes with different length arms (Supplementary Fig. 1h) may be due to the

centromere insulating the process that leads to differences between arms. In agreement with this, compaction is interrupted at centromeres in Hi–C maps (Fig. 1a, Supplementary Fig. 2b).

**Rec8-dependent punctate interactions emerge in meiosis.** Zooming in to consider within-arm organisation revealed punctate grid-like Hi–C interactions between pairs of loci during prophase (Fig. 2a), particularly prominent in *ndt80Δ* (Fig. 2a, b). Indeed, the focal meiotic patterns we observe resemble peaks between CTCF sites[31] rather than TADs[32,33] detected in mammalian interphase Hi–C maps, and likely arise from a heterogeneous mixture of 'transitive' interactions and 'skipping' of peak bases (Fig. 2c).

Genomic regions underlying the punctate Hi–C interactions display a remarkable visual (Fig. 2a, b), and quantitative (Fig. 2d–g), correspondence with previously characterised sites of high Rec8 occupancy[30]. A reciprocal analysis of calling Hi–C peaks and assaying the frequency of Rec8 sites around peak anchors confirmed this correspondence (Supplementary Fig. 4a). At pachytene, Rec8 sites display elevated cis/total contact frequencies (Fig. 2d), enriched contact frequency (Fig. 2e, f), and evidence of insulation (Fig. 2g)—features that correlate with Rec8 occupancy measured by ChIP (Fig. 2a, lower) consistent with recent observations[20]. In wild-type cells, Rec8-Rec8 interactions became visible in early prophase (2 h), peaked at mid prophase (4 h), and were especially prominent in the homogenous *ndt80Δ* cell population (Fig. 2a, b, f, Supplementary Fig. 4b, c). Importantly, Rec8-Rec8 enrichments are strongest between adjacent sites, decrease between non-adjacent sites with increasing genomic separation, and are absent in *trans* (Supplementary Fig. 4b, c). As for enrichments between CTCF sites in mammalian interphase[34], these observations argue that a cis-acting process generates such focal interactions in meiosis.

Rec8 is a central component of the meiotic chromosome axis[4]. *rec8Δ* mutants fail to assemble chromosome axes as detected by EM, and exhibit delayed and inefficient chromosome segregation producing few viable spores[4]. Assaying a *rec8Δ* mutant in the *ndt80Δ* background enabled us to determine that Rec8 is absolutely required for the emergence of the grid-like Hi–C patterns present in meiosis (Fig. 2a, b). Moreover, *rec8Δ* cells completely lose the shoulder in *P(s)*, indicative of a dramatic loss of arm compaction (Fig. 2b, Supplementary Fig. 4d), similar to that caused by depletion of SMCs in diverse contexts[24,26,28,35–38]. Instead of assembling an axis of loops, *rec8Δ* cells appear to be caught in a state with highly clustered telomeres (Supplementary Fig. 3, Supplementary Fig. 4e), consistent with previous observations by microscopy[39,40]. Moreover, in *rec8Δ* cells *cis* contact frequency is reduced (Fig. 2d, Supplementary Fig. 2c), similar to G1 cells, and cis/total no longer correlates with Rec8 occupancy. Instead, *rec8Δ* cis/total displays a decreasing trend along chromosome arms (Supplementary Fig. 4e), likely due to persistent telomere clustering (Supplementary Fig. 3a). Importantly, because focal interactions in wild-type cells are detected as early as cells start progressing through S phase (Fig. 2a, 2 hours), the lack of such interactions in *ndt80Δ*-arrested *rec8Δ* cells, which have completed DNA replication (Supplementary Fig. 2d), cannot be explained by partial arrest prior to a pachytene-like stage.

**Meiotic chromosomes modelled by loop extrusion with barriers.** To test how compaction and grid-like interaction patterns could jointly emerge in meiosis, we developed polymer simulations (Fig. 3a, Methods) similar to those used to successfully describe the assembly of TADs in mammalian interphase chromosomes[16]. Importantly, these simulations employ the *cis*-acting process of loop extrusion, where extruders form progressively larger chromatin loops, unless impeded by adjacent extruders or

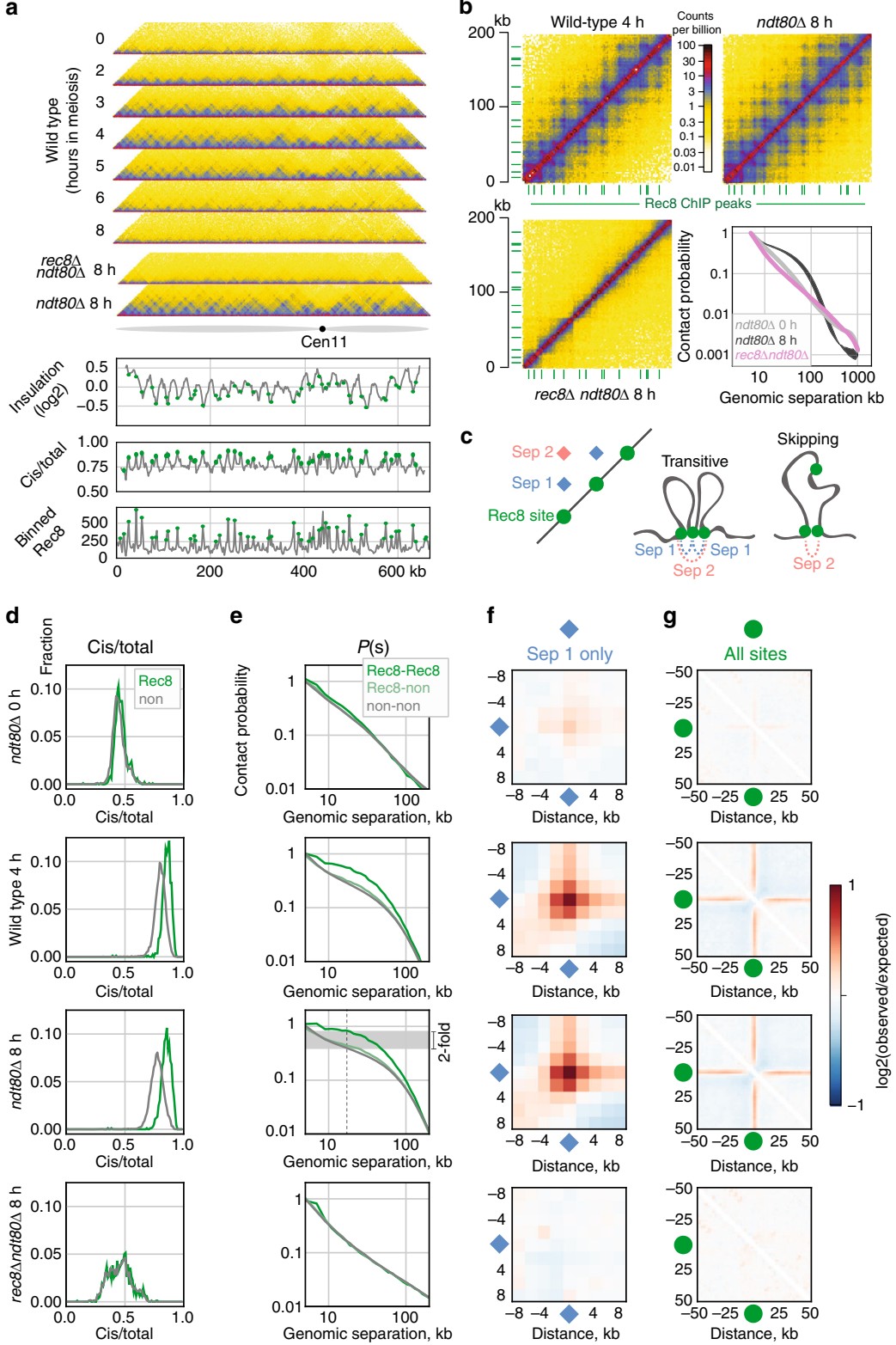

barrier elements (Fig. 3a). Extrusion dynamics are controlled by parameters dictating the processivity (average loop size) and separation (number of active extruders), as well as the strength of barriers (Methods). Because the accumulation of Rec8 at ChIP-seq sites[30] is indicative of barriers to extrusion[34], we positioned bi-directional barriers at Rec8 sites.

To find loop extrusion dynamics in agreement with experimental data, we computed the goodness-of-fit between experimental

*ndt80Δ* Hi–C maps and simulated Hi–C maps generated for a wide range of parameter combinations (Fig. 3, Methods). Models with excellent fits were identified in which ~64% of the genome is covered by extruded loops (Fig. 3b, c)—a far denser array than present in *S. cerevisiae* mitosis[28], but still less compact than human mitotic cells[27]. Even though extrusion can generate compaction independently of barriers (Fig. 3d, iii), an intermediate barrier strength is essential to match the grid-like patterns observed

**Fig. 2** Emergence of a Rec8-dependent grid of punctate interactions in meiosis. **a** Hi-C contact maps of chromosome 11 for the indicated genotypes at 2 kb resolution, showing near-diagonal interactions. *Lower panels*: log2(insulation); cis/total ratio, Rec8 ChIP-seq[30]. Insulation and cis/total calculated from *ndt80Δ* maps. Green circles: positions of Rec8 sites. Genome-wide cis/total (Spearman's $R = 0.62$, $P < 1e$-10) and insulation ($R = -0.23$, $P < 1e$-10, insulation window = 20 kb) profiles are correlated with Rec8 occupancy. Colour scale as in Fig. 2b. **b** Zoom-in of chromosome 11 (0–200 kb) for wt-4h, *ndt80Δ* and *rec8Δ*. Contact probability versus genomic distance, $P(s)$, for G1 (*ndt80Δ*-0h) and *ndt80Δ* and *rec8Δ*. Data are the average ($n = 2$) except for wt-4h. While faint locus-specific patterns exist in *rec8Δ*, there is no global enrichment at Rec8 sites (see **f** and **g**). Rec8 peak sites called from ChIP-seq data[30] are indicated in green. Interactive view: http://higlass.pollard.gladstone.org/app/?config=Twrh61jGT4SlxotaguTIJg. Comparison to published *ndt80Δ*-arrested Hi-C data[20]: http://higlass.pollard.gladstone.org/app/?config=NKoclcPJRTuah4ZrQPPm_Q. **c** Simplified illustration of how a grid of peaks on a Hi-C map can emerge between Rec8 sites either by transitive contacts between adjacent loops, or by loops that skip over adjacent sites. Experimentally observed grids extend much further than separation = 2 (Supplementary Fig. 4c). **d** Cis/total ratios for Rec8 (green) and nonRec8 (grey) sites for indicated datasets, showing an elevated cis/total frequency (0.85 versus 0.77) at Rec8 sites in *ndt80Δ*. **e** Contact probability versus genomic distance, $P(s)$, between Rec8-Rec8 sites (green), Rec8-nonRec8 sites (light green) and nonRec8-nonRec8 sites (grey). Note elevated pairwise contact frequency (~2-fold at 20 kb) at Rec8 sites in *ndt80Δ*. **f** Log2 ratio of contact frequency between adjacent Rec8 sites (separation = 1) compared to average *cis* interactions. **g** Log2 ratio of contact frequency centred at Rec8 sites compared to average *cis* interactions, showing mild insulation at Rec8 sites in *ndt80Δ*. These distinctions (**d–g**) are similar in wild-type pachytene (4 h) yet absent in G1 (*ndt80Δ*-0h) or in *rec8Δ*

experimentally (Fig. 3d, i). Despite the simplifying assumptions, simulated chromosomes displayed many features observed experimentally: (i) chromosomes fold into a loose polymer brush[4,41,42], with a Rec8-rich core[4] (Fig. 4a, Supplementary Fig. 5a, b); (ii) a grid-like interaction pattern naturally emerges in simulated Hi-C maps (Fig. 3d); (iii) importantly, because loop extrusion is a *cis*-acting process, pairs of Rec8 sites at increasing separations naturally have lower contact frequency (Fig. 3e).

Simulations also highlight the stochasticity of loop positions in the best-fitting models, with most barriers (73%) unoccupied by an extruder, and extruders paused with barrier elements on both sides only a minority of the time (15%) (Fig. 4b–d). Because of this, the majority (65%) of extruded loops cross over Rec8 sites, consistent with an average loop size roughly twice the average distance between Rec8 ChIP peaks (26 kb versus 12 kb, Fig. 4e), and remarkably consistent with estimates made using EM (~20 kb[41]). Genome-wide simulations for these best-fitting parameters show that the majority of chromosomes display similar goodness-of-fit with meiotic Hi-C data as on chr13 (Supplementary Fig. 6). Most strikingly—despite the prominence of Rec8-dependent grid-like features in the experimental data (Fig. 2a, b)—our simulations indicate that Rec8 sites are not always occupied by extruding cohesins and thus are present at the meiotic chromosome core in only a subset of cells, as inferred previously[43]. Notably, when loop extrusion operates independently on each chromatid, as in our simulations, the positions and sizes of loops are naturally heterogenous, even between sister chromatids (Fig. 4d). Such heterogeneity agrees with a recent microscopy study in *C. elegans* which argues for asymmetric chromatin loops on sister chromatids in meiosis[44].

The range of loop extrusion parameters we explored encompasses the situation where Rec8 sites always halt extrusion and *cis*-loops are formed between each consecutive Rec8 site. However, simulations with these parameters have quantitatively poor fits with experimental maps (Fig. 3d–e, ii): the bend in $P(s)$ comes too early to recapitulate experimental $P(s)$, and Rec8-Rec8 contacts are much too strong. The poor fit of such 'direct-bridging' simulations underscores the conclusion that only a fraction of Rec8 sites are occupied in a given cell, and argues that cohesin-dependent *cis*-loops must link regions that are not primary Rec8 binding sites in order to provide compaction without making Rec8-Rec8 enrichments overly strong. As expected, certain loop extrusion parameter sets give rise to TAD-like patterns. However, simulations with TAD-like patterns show poor quantitative agreement with experimental *ndt80Δ* data (Fig. 3d–e, iv), arguing that the patterns we observe in meiosis are better described as grids-of-peaks rather than a segmentation into TADs, and underscoring how a single process, loop extrusion limited by barriers, can give rise to multiple distinct 3D contact patterns.

A crucial prediction of our loop extrusion simulations is that depletion of extruders in meiosis would lead to both decompaction (Supplementary Fig. 5a–c) and loss of the grid-like pattern of Hi-C interactions. When we repeated our fitting procedure for *rec8Δ*, the best fits were for simulations with either no, or very few, extruded loops (Supplementary Fig. 5e). The lack of compaction in these simulations is consistent with previous EM showing decompacted chromatids in *rec8Δ*[4]. Such joint consistency between Hi-C and imaging data further supports loop extrusion as a mechanism underlying assembly of the cohesin-rich core and contributing to chromosomal compaction in meiosis. Our simulations also open the possibility that overly shortened axes observed upon Wapl[45,46] and Pds5[47] depletion may reflect heightened extruder processivity[48] upon which shortened SCs are assembled, and predict that such perturbations would cause a rightward shift in the $P(s)$ shoulder measured via Hi-C (Supplementary Fig. 5c).

**The synaptonemal complex modulates chromosome compaction.** To investigate how homologue synapsis affects chromosome conformation, we assayed pachytene cells in the absence of Zip1, the transverse filament of the SC[5], and Hop1, an axial element required for Zip1 loading[7] (Fig. 5a, b). Despite unsynapsed axial elements forming[4,5], *hop1Δ* and *zip1Δ* mutants proceed through both meiotic nuclear divisions—with a partial delay in prophase I in *zip1Δ* cells—generating spores with low viability[49–51]. Thus, to aid direct comparisons we again combined the use of the *ndt80Δ* allele to prevent exit from prophase. Both *zip1Δ* and *hop1Δ* retained punctate Hi-C interactions (Fig. 5b, Supplementary Fig. 4b, c), and displayed compaction relative to G1 or *rec8Δ*, but with the $P(s)$ shoulder shifted left relative to *ndt80Δ* (Fig. 5c). Attempts to model the known *zip1Δ* and *hop1Δ* defects in chromosome synapsis simply by removing interhomologue crosslinks from best-fitting *ndt80Δ* simulations did not recapitulate the $P(s)$ shift observed experimentally (Supplementary Fig. 5f). Instead, best-fitting simulations had shifts towards slightly lower processivity and larger separation (Fig. 5d), consistent with less axial compaction relative to the *ndt80Δ* control (Fig. 3c). Interestingly, subtelomeric regions no longer displayed a distinct $P(s)$ in *zip1Δ* and *hop1Δ* (Fig. 5e), suggesting that chromosome compaction at chromosome termini is regulated differentially.

## Discussion

Our analysis of meiotic chromosome organisation via Hi-C reconciles the function and localisation of factors thought to shape meiotic chromosomes with their 3D organisation (Fig. 6a), revealing the emergence of a punctate grid of interactions

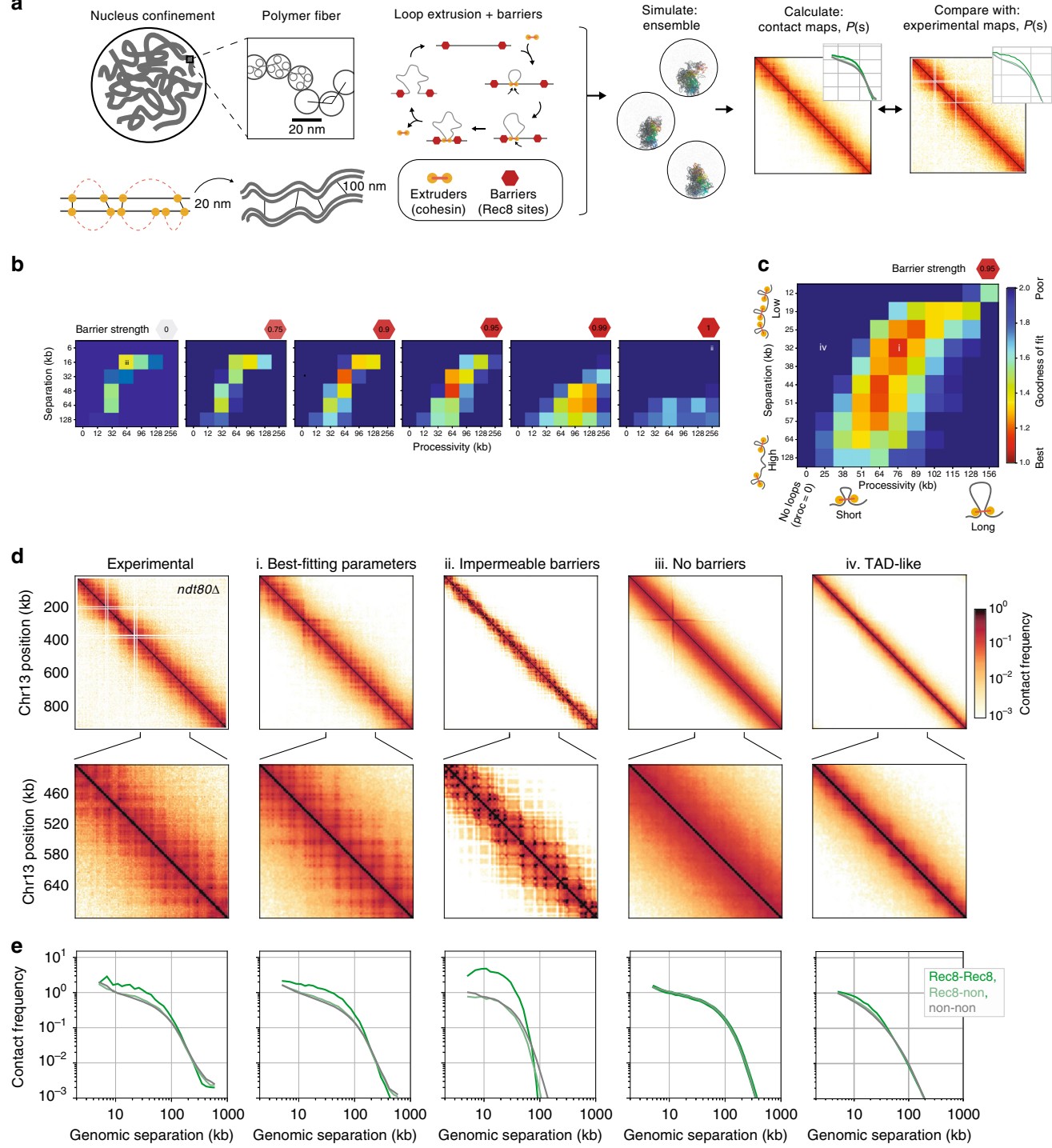

**Fig. 3** Modelling meiotic chromosome compaction. **a** In simulations, yeast Chr13 was represented as a polymer fibre confined to the nucleus subject to additional meiosis-specific constraints. These include extruded loops, sister crosslinks and homologue crosslinks (Methods). Barriers to extruded loops were placed at Rec8 sites[30]. We imposed inter-sister and inter-homologue crosslinks at sites of extruded loop bases in order to approximate the paired arrangement of homologues at pachytene. For each set of extruded loop parameters (processivity, separation and barrier strength), conformations were collected and used to generate simulated contact maps. Roughly, processivity dictates the size of an extruded loop unimpeded by collisions, separation controls the number of active extruders on the chromosome, and barrier strength controls the probability that an extruder gets paused when attempting to step past a barrier. Goodness-of-fit was then evaluated using the combined average fold discrepancy between P(s) curves for Rec8-Rec8, Rec8-non and non-non bin pairs at 2 kb resolution. Note that a value of 1 indicates perfect agreement between simulations and experimental data. **b** Goodness-of-fit for indicated barrier strengths over coarse grids of processivity and separation demonstrate that intermediate barrier strengths are required to agree with experimental *ndt80Δ* Hi-C maps. **c** Goodness-of-fit for a fine grid of processivity versus separation at barrier strength 0.95 (and for 0.90; Supplementary Fig. 5d). Best-fitting models had separation ~32 kb and processivity ~76 kb, corresponding to ~64% coverage of the genome by extruded loops of average length 26 kb. **d**. From left to right: contact maps for chromosome 13 for experimental data, and simulations with (i) best-fitting parameters, (ii) relatively stable loops between neighbouring Rec8 sites, (iii) no barriers, (iv) square TAD-like patterns of enriched contacts. Positions for each of these parameter sets indicated in **b**, **c**. **e** P(s) split by Rec8-Rec8, Rec8-non and non-non, as in Fig. 2e

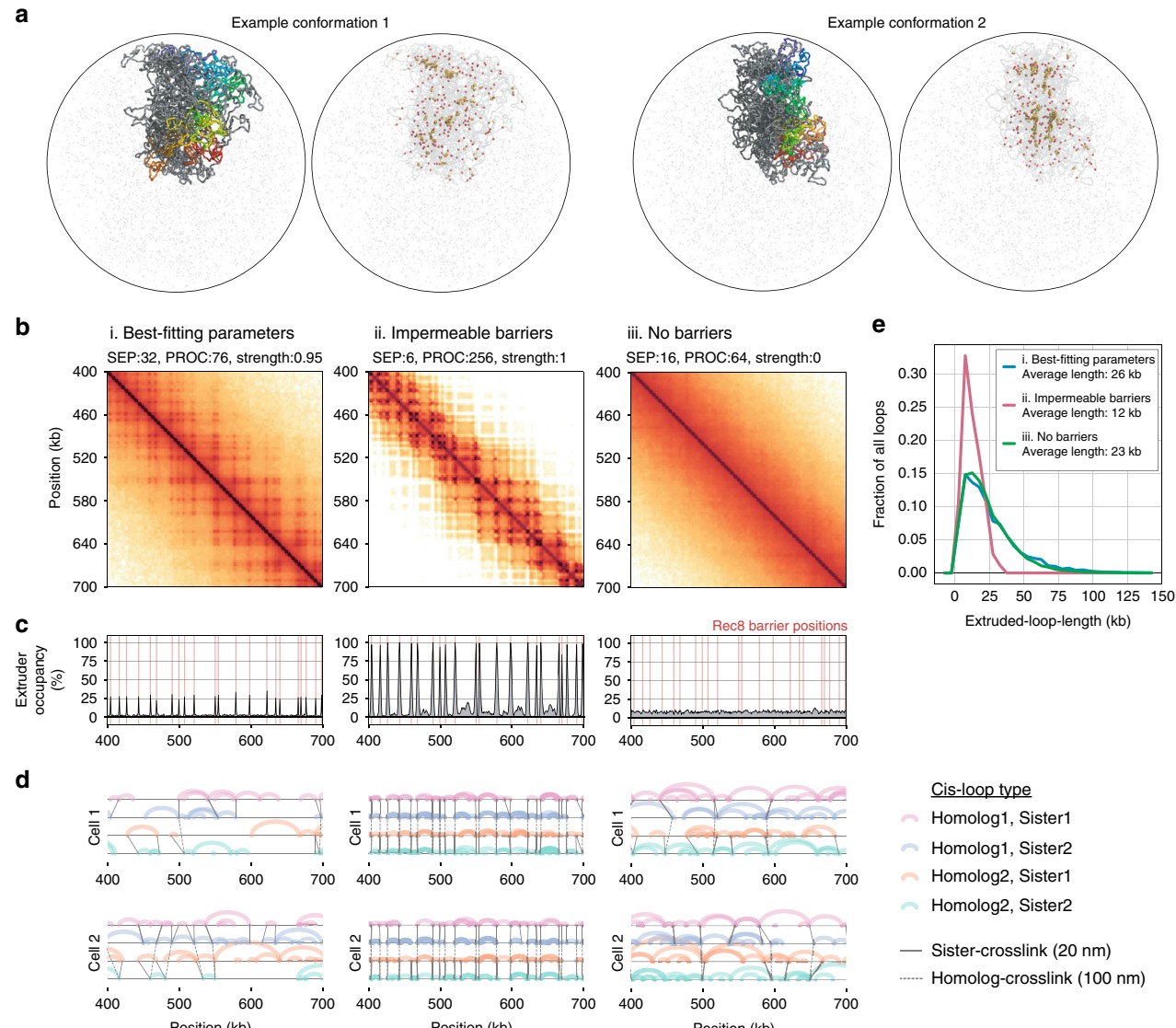

**Fig. 4** Polymer simulations of loop extrusion with varying barrier strength elucidate the heterogeneity of meiotic loops. **a** Two example conformations (top and bottom) for best-fitting simulations, which highlight: (left) one chromatid coloured from start (red) to end (blue); (right) extruders (yellow), extrusion barriers (red) and extruders paused at barriers (orange). **b** Simulated contact maps for the indicated region of chromosome 13 for: (i) best-fitting simulations, (ii) simulations with relatively stable loops between neighbouring Rec8 sites (barrier strength = 1 and high processivity), and (iii) no barriers, as in Fig. 3d. **c** Simulated ChIP-seq profiles for the indicated region of chromosome 13. Best-fitting simulations (i) display occupancy well below 100% at Rec8 sites. Simulations with stable loops (ii) display highly occupied Rec8 sites. Simulations without barriers (iii) have homogenous Rec8 occupancy across the genome. **d** Positions of extruded loops (arcs) sister crosslinks (solid black lines) and homologue crosslinks (dashed lines) for four chromatids in two separate cells, showing how the simulated Hi–C maps and ChIP-seq profiles emerge from the stochastic positioning of extruded loops from cell-to-cell. **e** Histogram of extruded loop lengths for indicated parameters (i, ii, iii)

concomitant with initial stages of meiotic chromosome compaction. Crucially, we formally demonstrate the link between preferential positioning of meiotic cohesin along the genome[11,30] and the inference that these loci come into close proximity based on the localisation of Rec8 to the chromosomal axes[4]. In the context of recent characterisations of mammalian meiosis via Hi–C[17–19], our results highlight similarities and differences across species. In all cases, meiotic contact frequency versus distance curves display a prominent shoulder consistent with *cis*-loop formation. Only *S. cerevisiae*, however, display punctate grid-like patterns of Hi–C enrichment all along chromosomal arms. This argues that the positioning of underlying loops may be much more stochastic from cell-to-cell in mammalian meiosis. Additionally, the plaid-like patterns observed in mammalian meiosis,

yet not observed in yeast, suggest that additional mechanisms, beyond loop extrusion, are at play in mammalian meiosis.

Remarkably, the punctate cohesin-dependent interactions in yeast meiosis emerge despite the absence of CTCF in this organism; this challenges previous models where focal Hi–C peaks are strictly dependent on CTCF[31,37,52], and indicates that alternative mechanisms of loop positioning must exist. Transcription constitutes a promising candidate for a mechanism of loop positioning that does not rely on CTCF[53–55]. Indeed, previous studies highlight the correspondence between cohesin positioning and convergent transcription in both yeast meiosis[11] and mitosis[56,57]. Moreover, whilst much less prominent than in meiosis, we find that locus-specific folding is evident in new high-resolution Hi–C maps of mitotic cells (Fig. 6b, c). Finally, in

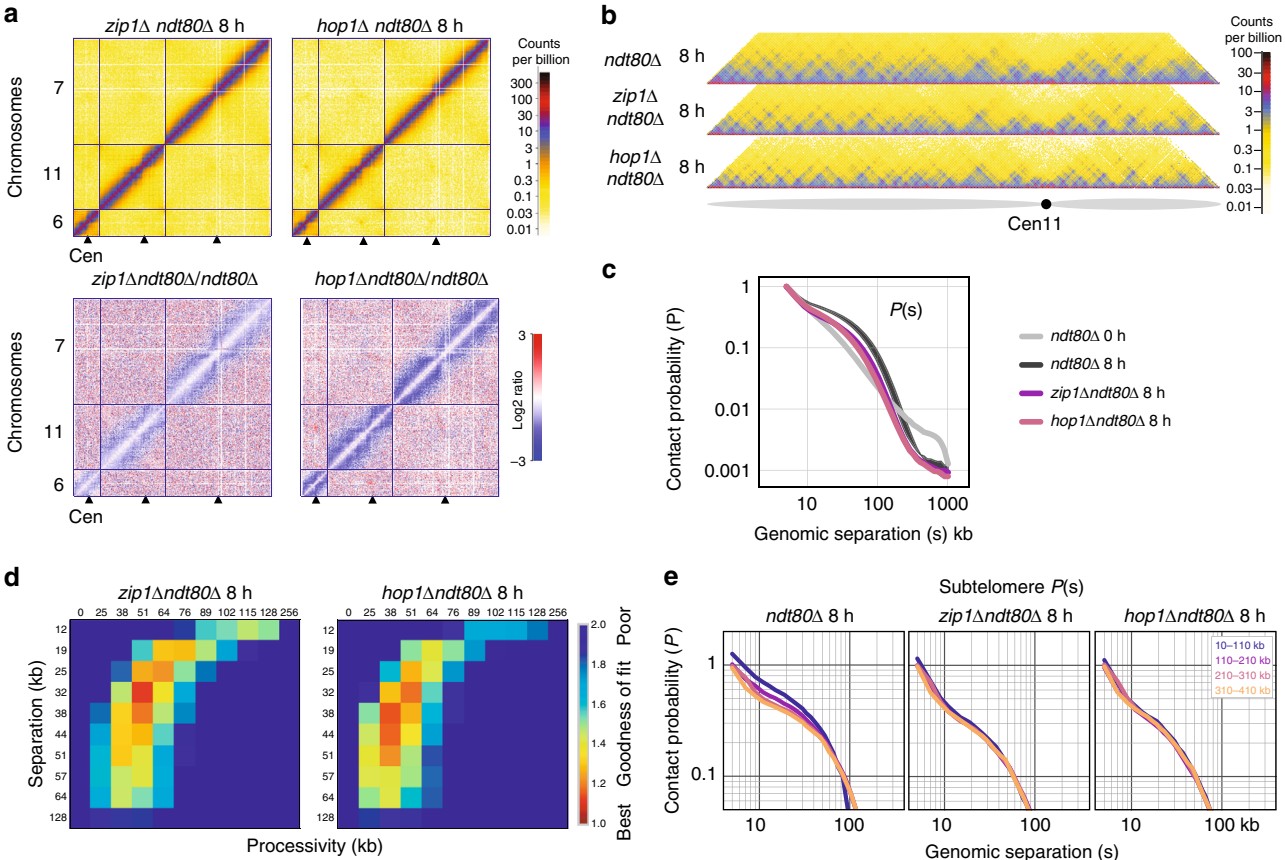

**Fig. 5** Hop1- and Zip1-dependent compaction of Rec8-dependent loops. **a** Top: Hi-C maps for *hop1Δ* and *zip1Δ* (plotted as in Fig. 1a). Bottom: Log2 ratio of *hop1Δ* or *zip1Δ* over control (as in Fig. 1g). For interactive views of the full genome, see: http://higlass.pollard.gladstone.org/app/?config=TTBGu5DDR0SHAa09zrjTXA. **b** Hi-C contact maps of chromosome 11 for *hop1Δ* and *zip1Δ* plotted at 2 kb bin resolution, showing near-diagonal interactions, as in Fig. 2a. **c** Contact probability versus genomic distance, *P(s)*, for G1, *ndt80Δ*, *hop1Δ*, *zip1Δ*. Shaded area bounded above and below by *ndt80Δ* replicas. Average between two replicas for *zip1Δ* and one sample for G1 and *hop1Δ* are shown. **d**. Goodness-of-fit for simulations without homologue crosslinks with a fine grid of processivity versus separation at barrier strength 0.95 *zip1Δ* and *hop1Δ*. **e**. Contact probability over genomic distance, *P(s)*, averaged over all chromosome arms stratified by distance from the telomere

agreement with a role for transcription in loop positioning, we observe an enrichment of convergently oriented TSSs around both meiotic and mitotic peak anchors (Fig. 6d, e).

However, whether or not it is transcription per se or the binding of large protein complexes-like RNA polymerase that influences loop positioning is unclear. We favour the view that transcription-associated machinery acts as a barrier to cohesin-dependent loop extrusion (Fig. 6a), rather than as a motive force as previously proposed[11,58,59], consistent with transcription-independent compaction by cohesin in mammalian interphase[60] and direct observation of extrusion by the related SMC condensin in vitro[61]. Indeed, the fact that chromosome compaction is interrupted at centromeres in both meiosis (Fig. 1a, g) and mitosis[54,62,63] supports the concept that large protein complexes—like the kinetochore or RNA polymerase—act as potent barriers to loop extrusion.

The reason for why loops are more prominent and strictly positioned in meiosis compared to mitosis is intriguing. However, our observations enable us to rule out the axial element, Hop1, the SC lateral element, Zip1 and the process of homologous recombination mediated by Spo11, Sae2 and Dmc1 (unpub. obs.) as important for the generation of such patterns. The axial element Red1, however, localises to chromosome axes prior to Hop1 in budding yeast[6], and the additional observation that Rec10 (ScRed1) binds to cohesin in fission yeast meiotic prophase[64] make Red1 a possible candidate for regulating loop

extrusion dynamics. The strong meiotic Hi-C patterns are also reminiscent of the grid-like Hi–C patterns observed in interphase mammalian cells upon depletion of the cohesin unloader, Wapl[37,60], wherein "vermicelli"-like chromatids arise with a cohesin-rich backbone[65]—emphasising the influence of cohesin dynamics on loop extrusion, and suggesting that differential cohesin regulation might underpin the differences between meiosis and mitosis.

Exploring our Hi–C data via polymer simulations enabled us to reveal a nuanced picture of meiotic chromosome assembly: loops are, on average, larger than the inter-Rec8 peak distance, and more than half of the loop bases are not associated with preferred sites of Rec8 binding. It is likely that loop sizes and positions vary widely from one cell to another, making classifications of genomic regions as 'axis' or 'loop' a great oversimplification. Our simulations also illustrate how a single mechanism—loop extrusion—can give rise to divergent Hi-C patterns, either TADs or grids-of-peaks, relevant in different cellular contexts. Indeed, simulations allow us to quantitatively test mechanisms of chromosome folding in addition to reporting the patterns observed in Hi–C maps. Looking more broadly, the agreement between our simulations and experimental data furthers the case for loop extrusion as a general mechanism[20,26–28,34,66–69] that is flexibly employed and regulated in interphase, mitosis and meiosis.

Finally, our results also reveal how the interplay between the synapsis components, Hop1 and Zip1, influences chromosome

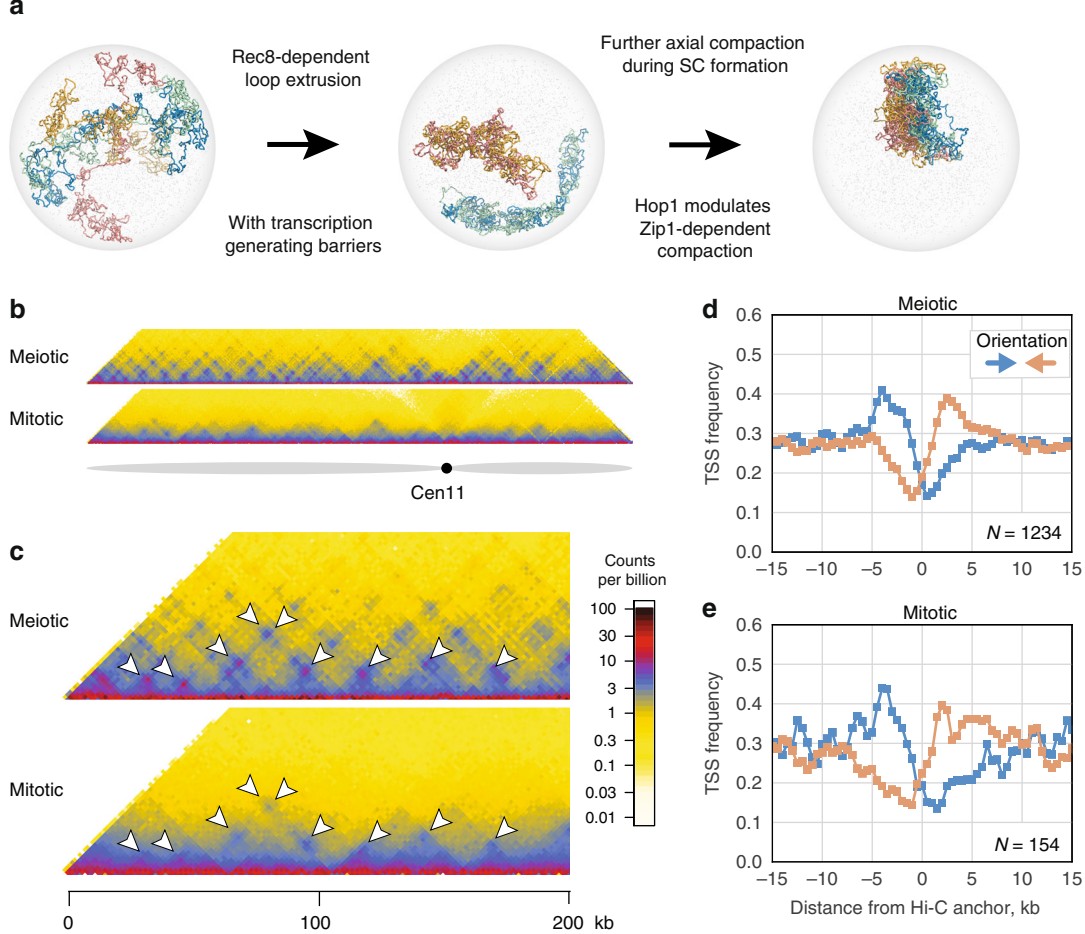

**Fig. 6** Underlying mechanisms of chromosome conformation in meiosis and mitosis. **a** Pathway of meiotic chromosome compaction: Rec8-dependent loop formation leads to initial chromosome arm compaction and emergence of a grid-like pattern of Hi-C interactions that jointly agrees with a mechanism of loop extrusion including barrier elements. We suggest that transcription could impose such barriers. Hop1 and Zip1 are dispensable for this step, but are required for synapsis, where additional compaction occurs differentially along chromosome arms. **b** Hi−C contact maps of chromosome 11 for meiotic (*ndt80Δ*, pachytene - top) and mitotic (wild type, nocodazole arrest - bottom) plotted at 2 kb bin resolution. **c** Zoom-in into contact maps on chromosome 11 (0–200 kb) of *ndt80Δ* (top) and mitotic (*bottom*). Arrowheads indicate sites of visually prominent focal interactions. **d**, **e** Frequency of TSSs by orientation around meiotic (**d**) and mitotic (**e**) Hi−C peak anchors, in 500 bp bins smoothed with a sliding window of three bins to emphasise the enrichment patterns (Methods). N = number of peaks analysed

morphology. That Hop1 and Zip1 are both required to increase chromosome compaction at pachytene likely points at their joint role in promoting synapsis[5,7], and supports the view that synapsis itself modulates axial compaction (Fig. 6a). While mouse spermatocytes defective for SC formation also show changes in chromosome compaction[17], developmental arrest of these mutants in a zygotene-like state makes it difficult to interpret the contribution of the SC to this phenotype. Our data suggests that the SC does have an impact on chromosome compaction because even though *zip1Δ* cells are partially defective in exiting prophase I, *hop1Δ* cells are not[49–51]. Interestingly, whilst Zip1 binds largely uniformly along the arms of pachytene chromosomes[70], subtelomeres and short chromosomes display an increase in short-range contacts and an earlier shoulder in *P(s)*, consistent with smaller loops or less compression of spacers between loops in these regions, and therefore less axial compaction. Because such differences correlate with disproportionate retention of Hop1 in these regions[70] and diminished efficiency of synapsis[71], it is possible that Hop1 impedes the pathway whereby Zip1 imposes additional compaction upon synapsis. Nevertheless, it is unclear whether Zip1 mediates this effect by modifying loop extrusion dynamics, or via a distinct process of axial compression, as has

been argued for higher eukaryote mitotic chromosome compaction[27]. Given the influence that chromosome structure has over so many aspects of meiosis, teasing apart these mechanisms is of great future interest.

## Methods

**Yeast strains and cell culture growth**. Strains used in this study were derived from SK1 and are listed in Supplementary Table 1. Key genes of interest are summarised in Supplementary Table 2.

**Monitoring DNA replication and nuclear divisions**. Cells were fixed in 70% EtOH, digested with 1 mg/ml RNAse (10 mM Tris-HCl pH 8.0, 15 mM NaCl, 10 mM EDTA pH 8.0) for 2 h at 37 °C, 800 rpm in Thermomixer (Eppendorf) and subsequently treated with 1 mg/ml Proteinase K in 50 mM Tris-HCl pH 8.0 at 50 °C, 800 rpm (as above) for 30 min for analysis by FACS. Cells were then washed in 50 mM Tris-HCl pH 8.0 and stained in the same buffer with 1 μM Sytox green or 1 μg/ml Propidium Iodide (PI) overnight in the fridge. Samples were processed on an Accuri C6. Collected FACS profiles were plotted with R using the library *hwglabr2* (https://github.com/hochwagenlab/hwglabr2), applying the following gates: For Sytox green (gate = c (200000, 3000000)) and for PI (gate = c (800, 10000)). Fixed cells were also used for quantification of nuclear divisions by spreading onto a microscope slide, mounting with Fluroshield containing DAPI followed by analysis with a Zeiss Scope.A1 microscope.

**Hi–C library preparation**. The Hi–C protocol used was amended from[72] by ~5-fold reduction in all materials and volumes. Briefly, for meiotic samples, *S. cerevisiae* diploid cells were synchronised in G1 by growth at 30 °C for ~16-18 h in 30 ml YPA (1% Yeast extract, 2% Peptone, 1% K-acetate) to OD600 of ~4, harvested, washed and resuspended in prewarmed sporulation medium (2% K-acetate with 0.2x nutritional supplements adenine, histidine, leucine, tryptophan and uracil) before fixing 5 ml aliquots (20–30 ODs) of relevant timepoints with formaldehyde at 3% final concentration for 20 min at 30 °C, in an orbital shaker at 250 rpm, then quenched by incubating with a final concentration of 0.35 M Glycine (2x the volume of Formaldehyde added) for an additional 5 min. Cells were washed with water, split into two samples (for two libraries) and stored at −80 °C ready for library preparation. For mitotic samples, *S. cerevisiae* diploid cells were grown in YPD (1% Yeast extract, 2% Peptone, 2% Glucose) to exponential phase, 10 μg/ml of nocodazole were added and 100 ml of cells (50–80 OD, sufficient for 1 Hi-C library) were fixed and stored (as described above). Cells were thawed, washed in spheroplasting buffer (SB, 1 M Sorbitol, 50 mM Tris pH 7.5) and digested with 100 μg/ml 100 T Zymolyase in SB containing 1% beta-Mercaptoethanol for 15–20 min at 35 °C. Cells were washed in restriction enzyme buffer (NEB3.1), chromatin was solubilised by adding SDS to 0.1% and incubating at 65 °C for 10 min. Excess SDS was quenched by addition of Triton X100 to 1%, and chromatin was incubated with 2.07 U/μl of *Dpn*II overnight at 37 °C. DNA ends were filled in with nucleotides, substituting dCTP for biotin-14-dCTP using Klenow fragment DNA polymerase I at 37 °C for 2 h, followed by addition of SDS to 1.5% and incubation at 65 °C for 20 min to inactivate Klenow and further solubilise the chromatin. The sample volume was diluted 15-fold, crosslinked DNA ends ligated at 16 °C for 8 h using 0.024 U/μl of T4 DNA ligase, and crosslinks reversed by overnight incubation at 65 °C in the presence of proteinase K. DNA was purified by phenol:chloroform:isoamylalcohol extraction and precipitated with ethanol, dissolved in TE and passed through an Amicon 30 kDa column. DNA was further purified by phenol:chloroform:isoamylalcohol extraction and precipitated again before treating with RNase A at 37 °C for 1 h. Biotin was removed from unligated ends by incubation with T4 DNA polymerase and low abundance of dNTPs (0.05 mM) at 20 °C for 4 h and at 75 °C for 20 min for inactivation of the enzyme. DNA was subsequently fragmented using a Covaris M220 (Duty factor 20%, 200 cycles/burst, 350 s, 20 °C), purified with Qiagen MinElute columns and DNA ends were repaired using T4 DNA polymerase, T4 Polynucleotide Kinase and Klenow fragment DNA polymerase I. DNA was purified with Qiagen MinElute columns and A-tailed before isolating fragments of 100–250 bp using a Blue Pippin (Sage). Biotinylated fragments were enriched using streptavidin magnetic beads (C1) and NextFlex (Bioo Scientific) barcoded adaptors were ligated while the DNA was on the beads. Resulting libraries were minimally amplified by PCR and sequenced using paired-end 42 bp reads on a NextSeq500 (Illumina; Brighton Genomics).

**Hi-C data processing and analysis**. Hi-C sparse matrices were generated at varying spatial resolutions using the Hi-C-pro pipeline[73], using a customised S288c reference genome ('SK1Mod', in which high confidence SK1-specific polymorphisms were inserted in order to improve read alignment rates[74]) and plotted using R Studio (version 1.0.44) after correcting for read depth differences between samples. Raw read statistics are presented in Supplementary Table 3. Repeat biological samples gave broadly similar matrices and, unless indicated otherwise, were averaged to improve their expected quantitative accuracy. As visual inspection indicated a number of potential translocations in the SK1 strain as compared with the S288c reference genome, for conservative downstream analyses, additional bins were masked if they contained potential translocations. Such bins were identified if they either had values in *trans* at the level of the median of the third diagonal in *cis*, or the maximum value in *trans* exceeded the maximum value in *cis* for SSY14 for bins displaying these properties in either *ndt80Δ*-0h or in *ndt80Δ*−8h and for MJ6 in wt-0h or wt-4h. chr1 was excluded from downstream analysis as few informative bins remained after filtering potential translocations.

Average maps centred at centromeres and telomeres were calculated as in Hsieh et al.[75], ensuring that collected patches for average centromere maps did not extend inter-chromosomally, and collected patches for average telomere maps did not extend beyond centromeres or inter-chromosomally. Contact frequency versus distance curves, *P(s)*, were calculated from 2 kb binned maps, with logarithmically-spaced bins in *s* (numutils.logbins, https://bitbucket.org/mirnylab/mirnylib, start = 2, end = max(binned arm lengths), $N = 50$), and restricting the calculation to bin pairs within chromosomal arms and excluding bins less than 20 kb from centromeres or telomere (as in Hsieh et al.[75]), and normalised to the average value at 4 kb. *P(s)* stratified by distance to telomeres was calculated using the combined distance to telomeres for each bin-pair (as in Mizuguchi et al.[76]), and excluded bins-pairs where one bin was closer to a centromere than telomere along that arm. Distance to centromeres, and *P(s)* stratified by this distance, was calculated similarly. Log2 insulation profiles were calculated using a sliding diamond window (as in Crane et al.[77]) with a ± 20 kb ( ± 10 bins) extent; as in Nora et al.[52] downstream analyses were restricted to when there were zero or one filtered bins in the sliding window. To calculate histograms of cis/total (Fig. 2d), bins were defined as either Rec8 or non-Rec8. To calculate *P(s)* split by Rec8 bin-pair status, each bin-pair (i.e. entry of the heatmap) was assigned as either Rec8-Rec8, Rec8-nonRec8 or non-non (e.g. Figure 2e). *P(s)* was then aggregated separately across chromosomes for these three categories, similar to calculation of *P(s)* within and between TADs[34].

Average log2 observed/expected maps were calculated by first dividing by intra-arm *P(s)* and then averaging together appropriate patches of Hi-C maps. Correlations between Rec8 occupancy from[30] and insulation or cis/total profiles excluded chromosome 12 because the rDNA locus greatly alters the insulation profile within the right arm of the chromosome.

For display via HiGlass[1], Hi-C matrices were processed with the distiller pipeline (https://github.com/mirnylab/distiller-nf) and stored in cooler format (https://github.com/mirnylab/cooler[78]) compatible with HiGlass. Hi-C peaks were called using the call-dots command line tool in cooltools (https://github.com/mirnylab/cooltools), as recently employed for Micro-C XL data[79]. This identified locally enriched interactions (as defined previously[31]), for Hi-C data at 2 kb resolution (parameters:–dots-clustering-radius 4000–kernel-width 3–kernel-peak 1–max-loci-separation 200000–fdr .25). As the post-hoc filtering thresholds used in[31] appeared too stringent, we used a more relaxed FDR 0.25 and a 50% more lenient post-processing enrichment threshold (enrichment_factors of 1.25, 1.375, 1.5 instead of default 1.5, 1.75, 2.0 in thresholding_step).

**Polymer simulations**. Meiotic loop extrusion simulations begin with a generic polymer representation of the yeast chromatin fibre similar to that used in previous models of yeast mitotic chromosomes[28], where each 20 nm monomer represents 640 bp (~4 nucleosomes), confined to a 1 micron radius nucleus. We simulated the chromatin fibre with excluded volume interactions and without topological constraints, using Langevin dynamics in OpenMM[80,81], as previously[28]. This is achieved using a soft-core repulsive potential that allows for occasional chain crossing in steady-state. Importantly, meiotic simulations remove the geometric constraints specific to the Rabl conformation[82,83] because this is not visible in meiotic pachytene *ndt80Δ* Hi-C maps.

Because our focus was to characterise the grids of intra-chromosomal interactions, we considered a system with multiple copies of chromosome 13, equivalent to four copies of the haploid genome in terms of total genomic content (4 × 13 copies of chromosome 13), to enable efficient computational averaging of simulated Hi-C maps. Extruded loops were generated according to parameters that describe the dynamics of loop extruders, using the simulation engine as previously described[84]: extruder separation, extruder processivity, chromatin fibre relaxation time relative to extruder velocity, and barrier strength. Because yeast chromosomes are short compared to higher eukaryote chromosomes, relaxation time is relatively rapid and we focused on separation, processivity and barrier strength. At every given timepoint an extruded loop is realised as a bond between monomers at the two bases of the loop (see ./src/examples/loopExtrusion in https://bitbucket.org/mirnylab/openmm-polymer/).

Upon encountering a barrier, a loop extruder is paused with probability according to the barrier strength; barrier strength = 1 indicates an impermeable barrier, barrier strength = 0 indicates no impediment to extrusion. We assume loop extrusion occurs independently on each chromatid, and simulate loop extrusion dynamics on a 1D lattice (as previously described[34]) where the number of lattice sites equals the total number of monomers (75,140). Bi-directional barriers were placed at monomers with positions corresponding to Rec8 ChIP-seq sites[30], and pause extruders according the barrier strength parameter. We assume a uniform birth probability, constant death probability, and that all barriers have an equal strength; as additional data becomes available, these assumptions can be relaxed, and more detailed models can be built.

We investigated scenarios where chromatids are then either left individualised (52 copies), crosslinked to sisters (26 pairs), or additionally paired with homologues (13 pairs-of-pairs). For simulations with sister crosslinks, these were added (following Goloborodko et al.[68]) when extruded loop bases were present at cognate positions ±30 monomers (~20 kb) on both chromatids (distance = 20 nm); homologue crosslinks were added similarly when sister crosslinks were present on both chromatids (distance = 100 nm); centromeres and telomeres were always paired, and both presented impermeable (strength = 1) boundaries to extruders. To avoid introducing pseudo-knots, if extruded loops were nested only the outer cohesins were considered as possible bases for sister crosslinks, sister crosslinks were only allowed between the same side of loop bases (i.e. left-to-left arm or right-to-right arm), and sister crosslinks were only added between bases at the reciprocal minimum distance.

To calculate simulated Hi-C maps, contacts were recorded from conformations of the full system, which includes intra- and inter-sister, and interhomologue contacts. Because experimental Hi–C here does not distinguish either sisters or homologues, contacts were then aggregated into one simulated map. For each model and parameter set we investigated, we collected an ensemble of conformations at steady-state, generated simulated chromosome 13 Hi-C maps, and compared their features and *P(s)* with those from experimental Hi-C maps. Contacts were recorded between any two monomers in a given conformation separated by <60 nm. Each simulated chromosome 13 map represented an average over 5200 conformations. *P(s)* for chr13 was calculated from 2 kb binned simulated maps exactly as for experimental maps.

Goodness-of-fit between simulations and experimental data (e.g. Fig. 3b,c) was computed as the geometric standard deviation of the ratio of simulated to experimental *P(s)* combined across $P^{Rec8-Rec8}(s)$, $P^{Rec8-non}(s)$, and $P^{non-non}(s)$, as was previously done for *P(s)* within TADs of multiple sizes and between TADs[34],

for $s$ from 10 kb to 300 kb.

Goodness-of-fit

$$= \exp\left(1/N \sum_s \left( \begin{array}{l} \left(\log\left(P_{\text{sim}}^{\text{Rec8}-\text{Rec8}}(s)\right) - \log\left(P_{\text{expt}}^{\text{Rec8}-\text{Rec8}}(s)\right)\right)^2 + \\ \left(\log\left(P_{\text{sim}}^{\text{Rec8}-\text{non}}(s)\right) - \log\left(P_{\text{expt}}^{\text{Rec8}-\text{non}}(s)\right)\right)^2 + \\ \left(\log\left(P_{\text{sim}}^{\text{non}-\text{non}}(s)\right) - \log\left(P_{\text{expt}}^{\text{non}-\text{non}}(s)\right)\right)^2 \end{array} \right)^{1/2} \right)$$

Where $s$ indexes bins of increasing genomic distance, and $N$ is the number of bins. This measure reflects the typical fold-deviation in $P(s)$ of each of these three classes of bin pairs. Note that a value of 1 indicates a perfect agreement between simulations and experimental data.

Simulated ChIP-seq profiles (Fig. 4c) for Rec8 were generated by aggregating the position of extruded loop bases (two per extruded loop) across conformations. Statistics of extruded loop positioning relative to Rec8 sites was calculated with *loopstats.py* in *looplib* (https://github.com/golobor/looplib), and arc diagrams (Fig. 4d) with *loopviz.py*. Conformations showing chromatids or positions of extruded loop bases were rendered in PyMOL (https://pymol.org/sites/default/files/pymol.bib).

**Reporting summary**. Further information on research design is available in the Nature Research Reporting Summary linked to this article.

## Data availability

Raw sequence reads are accessible via the SRA repository GSE127940. Hi-C matrices publicly viewable via the interactive HiGlass viewer[1], hosted at http://higlass.pollard.gladstone.org. All other relevant data supporting the key findings of this study are available within the article and its Supplementary Information files or from the corresponding authors upon reasonable request. Data monitoring the cell culture— evaluation of meiotic progression by DAPI (Fig. 1c; Supplementary Fig. 1c) and FACS (Fig. 1b, f; Supplementary Fig. 1b; Supplementary Fig. 2d)—are available as a Source Data File. A reporting summary for this Article is available as a Supplementary Information file.

## Code availability

Code used to analyse Hi-C data publicly available online: https://bitbucket.org/mirnylab/mirnylib, https://github.com/mirnylab/cooltools. Code used to develop polymer simulations publicly available online: https://bitbucket.org/mirnylab/openmm-polymer/.

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

## Acknowledgements

We thank Tim Cooper for deploying the HiC Pro installation, Rachal Allison for critical reading of the manuscript, Scott Keeney and Franz Klein for sharing *S. cerevisiae* strains, Svetlana Lyalina for assistance with QB3 GPUs, Anton Goloborodko for suggesting the use of *looplib*, and Nezar Abdennur for feedback. SS and MJN are supported by an ERC Consolidator Grant (#311336), the BBSRC (#BB/M010279/1) and the Wellcome Trust (#200843/Z/16/Z). KP and GF are supported by NIMH grant #MH109907, NHLBI grant #HL098179, and the San Simeon Fund. J.B. is supported by BBSRC grant number BB/S001425/1.

## Author contributions

SAS and MJN planned the study, performed wet-lab work and data analysis. GF and KSP developed polymer simulations and performed data analysis. SAS, GF, JB, KSP and MJN discussed results and wrote the manuscript.

## Competing interests

The authors declare no competing interests.
