## [Peer Review File · Nature Communications]

Reviewers' comments:

Reviewer #1 (Remarks to the Author):

In this paper, the authors explored the meiotic chromosome structure via Hi-C in *Saccharomyces cerevisiae*. This study reconciles the localization and the related function of the factors that are thought to shape meiotic chromosomes and reveals the emergence of a punctate grid of interactions concomitant with meiotic chromosome compaction. Importantly, the authors demonstrated the punctate cohesin-dependent interactions in yeast meiosis emerge despite the absence of CTCF in this organism. This indicates that alternative mechanisms of loop positioning must exist. What's more, the author found that this grid-like interaction pattern emerges independently of meiotic chromosome synapsis; synapsis itself generates additional compaction that facilitate meiotic chromatin maturation. I have to say that this is one of the best and fun Hi-C papers I have ever read, and the data and the analysis are professional, clear, and convincing, which illuminates how meiotic chromosome forms in a mechanistic view. The paper was very well written and concise. I recommend for publication, provided if the authors can address the following suggestions.

Major comments:

1. It would be much helpful for the readers to understand the results if staining figures are available to visualize chromatin states at each stage, including for mutant cells.
2. Fig. 2, one important issue to clarify is that whether the Hi-C structure changes in *rec8* mutants are due to *Rec8* deficiency or the developmental arrest caused by the mutation. In the latter case, the Hi-C structure may reflect chromatin organization at the arrested earlier stage. I imagine that it could be a bit difficult to dissect these two possibilities.
3. Similarly, in the last part of the paper, the author concluded that the grid-like interactions emerge independently of meiotic chromosome synapsis complex. Will developmental defect or delay play a role in the differences between *Rec8* and *Zip1/Hop1* mutants?
4. The relationship between grid-pattern and transcription is interesting but not clearly demonstrated in the paper. It would be more convincing if the authors can provide analysis results of transcriptome data. Also, it would be interesting to explore whether the transcription is affected once the grid-pattern disappears, or vice versa.
5. Recently, several papers explored meiotic chromatin organization with Hi-C in mouse and monkey (Wang et al., *Mol. Cell*; Alavattam et al., *NSMB*; Patel et al., *NSMB*). It would be helpful if the authors can comment on the commonalities and differences between species.
6. The title may be somewhat general, especially considering the differences in meiotic chromatin assembly among different species.

Minor comments:

7. Extended data Fig. 2, why does the cen-cen show higher trans interaction in *ndt80* mutant yeast compared with WT at 0 and 4 hour?
8. Fig. 2g, should the lines be blue instead of red?
9. Fig. 4a-b, can the author also provide a statistical result for the *zip1* and *hop1* deficient cells, similar as Fig. 1d?
10. Some figure orders need to be fixed. Fig. S4 appeared ahead of Fig. S3. Fig. S4b-c appeared prior to S4a.

Reviewer #2 (Remarks to the Author):

In their manuscript, the authors describe the 3D genome organization of the yeast *S. cerevisiae* in meiosis by using Hi-C. They first observe general large-scale features consistent with previous knowledge such as general chromosome condensation and decrease in telomere interactions. On a finer scale, they observe focal interactions anchored at *rec8* sites, and show that these mostly disappear upon deletion of *rec8*. Next, they use polymer simulations to demonstrate that a loop extrusion mechanism with barriers placed at *rec8* sites can reproduce part of the observed data.

Finally, they show that mutants involved in homologue synapsis do not have a strong effect on the observed meiotic patterns.

Overall the paper is clear and both the experiments and analysis seem sound.

Comments:

1. Regarding novelty, there are several interesting results here but it is not clear enough that some of this has been done already: a previous paper by Muller et al Mol Syst Biol 2018 has measured yeast meiosis Hi-C (including the npt80 mutant which the authors here used), using a system of synthetic designed chromosomes which might even be considered superior for some of these questions. This previous work also showed the anchors at rec8 sites. The authors do cite this paper on the matter of rec8 sites. I suggest better clarifying in the text what has been already shown versus the new results.
2. It is unclear to me how goodness of fit is measured, and what the final quality of fit to the data is, for example in Figure 3d. It would also be helpful to provide an additional metric for the fit between these maps, e.g. Pearson correlation. Finally, it is important to test these metrics after normalizing the maps for genomic distance.
3. Following on the previous point, was there some kind of cross validation here? It is difficult for the reader to evaluate whether there is any overfitting here. To ensure there is no overfitting, it would be good to estimate parameters on one region and apply the model to a different region using these parameters and the rec8 sites for the new region.
4. Regarding the parameters of the model, if I understand correctly the authors re-estimate the parameters for the rec8 mutant. Doesn't it make more sense to keep the loop extrusion parameters the same?
5. It would be useful if the authors could provide a quantitative estimate of how synchronized the cells are, e.g. what fraction is in pachytene.
6. The authors suggest the rec8-anchored patterns are closer to focal interactions than to TADs. Looking at the figures this is not obvious, and it also seems somewhat different from the data of Muller et al. Is there a more quantitative way of making this claim, for example by 2d peak calling? At minimum I suggest showing a better figure of this. In any case, the distinction between the two cases does not seem that clear given that the underlying mechanism suggested here (loop extrusion) is the same mechanism used to explain TADs.
7. In Figure 2b, it seems like domain structure remains that is aligned with rec8 sites is present in the rec8 npt80 mutant. There seems to be no mention or explanation of this in the text.

Reviewer #3 (Remarks to the Author):

How chromosomes are condensed before cell division is an important issue in cell biology. It had been widely believed that condensin, rather than cohesin, plays a predominant role in chromosome compaction. However, recent studies suggest that topologically associated domains (TADs) of chromosome, which are captured as grid-like Hi-C interactions, are formed by extrusion of chromatin loops. This loop extrusion is largely mediated by cohesin and contributes to chromosome compaction in both interphase and mitosis in yeast. The authors analyzed grid-like Hi-C interaction of meiotic chromosomes in budding yeast. The data clearly indicate the essential role of Rec8 cohesin in assembling grid-like structure, which is not affected by meiosis-specific chromosome axis components. This study is novel and would be suitable for publication in Nature Communications if some concerns below are properly addressed.

Comments;

- 1) Line 62: Although telomere clustering is not extensive, bouquet formation is indeed conserved in budding yeast. Therefore, the description should be changed; e.g. telomere clustering was not detected.

2) Line 79-82: It is not clear whether only distinct behavior of subtelomeric and subcentromeric regions explains the different behavior of long and short arms. Namely, the short arm might be more influenced by them but not other unknown mechanism. Is this true?

3) Line 83-86: Regarding to the above point, the last two sentences of this section might be not significant.

4) Line 174-186, 212-219: A recent study shows that Rec8 cohesin complex directly bind Rec10 (Red1 homolog) in early prophase in fission yeast meiosis (Dev Cell 32, 220-230, 2015). Also in budding yeast, Red1 localizes at chromosome axis prior to Hop1 or other axis components (JCB 136, 957-967, 1997). Therefore, Rec8 cohesin complexes sitting at the barrier sites may be stabilized by Red10 (even without Hop1 and Zip1). This could be experimentally examined or mentioned as a possibility.

Reviewer #4 (Remarks to the Author):

The present manuscript by Schalbetter, Fudenberg and co-workers reports on extensive experiments and computer simulations to study the structural features of meiotic chromosomes in yeast. Interestingly, they provide independent confirmation for the decreasing of centromeric and sub-telomeric clustering during meiosis previously characterised by imaging, and substantial evidence that Rec8 is required for the emergence of the grid-like patterns in meiotic Hi-C maps. The computational analysis adds on the experiments providing robust evidence that the formation of this grid-like pattern is compatible with a mechanism of loop-extrusion promoted by convergent transcription with barriers localised at Rec8-enriched sites.

I find this work very relevant in helping our understanding of the meiotic chromosome structural organisation in yeast, but also in highlighting the underline folding mechanisms. I also appreciate the interdisciplinary approach used to tackle this challenging problem, which had to combine chromosome capture experiments (Hi-C) in wild-type and mutant yeast strains, and polymer simulations that properly allowed to extract useful mechanistic insights. Although I find that the manuscript has great potential and the results reported appear quite solid and well performed, the manuscript lacks a proper discussion of the existing literature and an adequate explanation of essential details of the simulation for the standards of Nature Communications. I recommend the authors to address the specific remarks mentioned here below.

Major points

1) The structure of the paper is not conventional for Nature Communications. In particular, the Introduction section is merged with the Results. This choice makes it difficult to easily distinguish the novelties of this work from the results in the literature. It would add to the clarity of the manuscript if the authors properly divide these two sections.

2) The division of Introduction and Results sections would help the authors to aptly comment on existing literature that is currently completely missing or poorly cited. Specifically, the authors should comment on the three recent publications (Alavattam et al., 2019, Nature Structural & Molecular Biology 26, 175–184; Patel et al., Nature Structural & Molecular Biology 26, 164–174 (2019), Wang et al., 2019, Molecular Cell 73, 547–561) that report on Hi-C experiments of meiotic chromosomes in three different species human, mouse and rhesus monkey respectively. Properly commenting on these published works would be beneficial for the community to put this work in the correct context and for the authors to fully highlight the novelties of their work.

3) The experiments and the simulations are quite sound to me. However, some details of the simulation protocol are currently missing. Addressing the following points would add on the reproducibility of their computational work:

- At page 20 line 541 the sentence "We simulated the chromatin fiber with excluded volume

interactions and without topological constraints" should be clarified in particular about the absence of topological constraints. Does it mean that the chains are phantom? In this case how this fits with the applied excluded volume interaction? Are the bonds crossable? Or does it mean that the chains are unknotted in the initial state? In this case, how do the authors check the initial unknotted state and its persistence along the entire simulation?

- What is the starting conformation of the simulated polymer chains? How does it affect the final result? I find this an important aspect, in particular, if the chains are not phantom.
- What is the size of the nucleus and, as a consequence, the density of the polymer solution? Is it chosen following experimental evidence?
- What contact radius has been used to compute the contact maps of the simulated systems?
- The visual similarity of the simulated (best-fitting parameters) vs experimentally measured contact maps is usually striking in the figures of the manuscript. However, when the authors compare the models with the experimental results, they rather prefer to examine the contact probability over the genomic distance ($P(s)$) that is somehow a coarser quantity. Although the comparison is still acceptable for validation, the authors should comment in the manuscript on this particular choice.

Some minor points need to be clarified or corrected. In particular:

- In page 2, line 58, Supplementary Fig. 2 should read Extended data Fig. 2.
- The panels in Figure 3c and 4e both represent the Goodness-of-fit, but their colour scales have different labels.
- The panels in Figure 2g and 4f are not referenced in the main text, and the Extended Data Figure 2c that is referenced in page 6 line 177 is missing.

Overview of revisions:

Below are our detailed replies to the reviewers' comments. In the manuscript, we have highlighted all text changes (in dark blue) as requested and are referring to them in the replies to reviewers directly and/or by line numbers. To integrate the reviewer's comments and suggestions, we have amended the structure of the manuscript by the following main changes:

- *Reassignment of Extended Data Figure 6 to a main Figure (Fig. 4) including example conformations of Figure 3:*
Emphasising our elucidation of the heterogeneity of the population of meiotic loops, an important aspect that we have highlighted in the revised abstract.
- *Addition of a new main figure (Fig. 6):*
Containing panels from the previous ED Figs. 4 and 7 and additional analysis of the correlation of transcription and loop anchor positioning
These analyses address the comments of reviewer 1
- *Addition of a new Supplementary figure 6:*
Genome-wide simulations (as requested by reviewer 2)
- *Separation of Introduction and Results sections:*
Including a new paragraph in the introduction and discussion addressing the relevant literature, published after submission of this manuscript
- *Subheadings in Results section as recommended*

We think these changes and further clarifications in the text in response to the reviewer comments and suggestions have improved our manuscript, and we are pleased to return our manuscript to the reviewers.

Detailed replies:

We thank the reviewers for their comments and for giving us the opportunity to clarify important aspects of our study. We answered each comment below and have modified the manuscript accordingly. In particular, we have introduced a separate introduction section describing the known chromosome morphology of the wild type and mutants we use. We have extended our simulations and analysis and added new figure panels to address the relevant questions. Finally, we updated and clarified the main text to discuss our findings in the context of recent literature. Please see below for detailed answers in which we refer to the relevant text changes and figures. In the manuscript, we have highlighted all text changes in dark blue. Together, we believe our manuscript has been improved thanks to the reviewers' suggestions and can be further considered for publication.

Reviewer 1:

Major comments

1. *It would be much helpful for the readers to understand the results if staining figures are available to visualize chromatin states at each stage, including for mutant cells.*

We agree that visualization can be helpful for the reader, particularly for a broad-interest journal. To address this point we added cartoons of the different stages (in **Fig. 1a top**). Unfortunately, we do not have any chromosome spreads from the exact samples we used for Hi-C, but we note that the genotypes we analyze have been well documented in the literature and we have added a short description of the reported meiotic chromosome morphology of each strain to the introduction and results sections to bring further clarification.

2. *Fig. 2, one important issue to clarify is that whether the Hi-C structure changes in *rec8* mutants are due to *Rec8* deficiency or the developmental arrest caused by the mutation. In the latter case, the Hi-C structure may reflect chromatin organization at the arrested earlier stage. I imagine that it could be a bit difficult to dissect these two possibilities.*

Thank you, this is a very good and important point. Whilst it is clear that *rec8* Δ cells have a prophase delay and deficiency in exiting prophase I, a substantial fraction of cells do undergo missegregation during the meiotic divisions resulting in aneuploidy (Klein et al., Cell, 1999). Because we are using the *ndt80* Δ background, at least a fraction of cells would have arrested at the *ndt80* Δ /pachytene stage. From our wildtype experiments we know that focal interactions start to appear as early as 2 hours (**Fig. 2a**), as soon as the first cells initiate DNA replication (**Fig. 1b**). We also know that the majority of the *rec8* Δ *ndt80* Δ cells have completed DNA replication (**Supplementary Fig. 2d**), which excludes the possibility that the lack of focal interactions is caused by a developmental arrest. To address this point we added more information on the morphology and phenotype of the mutants used in this study to the introduction and relevant results sections (lines 37-40, 123-125, 135-139).

3. *Similarly, in the last part of the paper, the author concluded that the grid-like interactions emerge independently of meiotic chromosome synapsis complex. Will developmental defect or delay play a role in the differences between *Rec8* and *Zip1/Hop1* mutants?*

As described above, we now discuss the morphological phenotypes of the mutants we are using in the introduction. Furthermore, we added a comment about the meiotic progression in the results section in which we are investigating *hop1* Δ and *zip1* Δ mutants (lines 210-213). In *S. cerevisiae* SK1 background, none of the three mutants (*rec8* Δ (Klein et al., Cell, 1999), *zip1* Δ (Sym and Roeder, Cell, 1994), *hop1* Δ (Carballo et al., Cell, 2008)) have a persistent developmental arrest phenotype. However, as noted above, *rec8* Δ (Klein, Cell, 1999) but also *zip1* Δ (Sym and Roeder, Cell, 1994) cells do exhibit a prophase I delay, which could result in a population of cells not reaching the *ndt80* Δ arrest stage. However, as the timing of meiotic progression in *hop1* Δ cells is similar to wild type (Carballo et al., Cell, 2008) and we do observe similar changes to chromosome compaction in these cells as we do in *zip1* Δ , our data suggests that these changes are not due to a developmental arrest. We have clarified this in the results section.

*“While, mouse spermatocytes defective for SC formation also show changes in chromosome compaction{Wang et al 2019}. However, developmental arrest of these mutants in a zygotene-like state makes it difficult to interpret the contribution of the SC to this phenotype. Our data suggests that the SC does have an impact on chromosome compaction because even though *zip1* Δ cells are partially defective in exiting prophase I, *hop1* Δ cells are not {Sym and Roeder 1994; Storlazzi 1996; Carballo 2008}.”*

4. *The relationship between grid-pattern and transcription is interesting but not clearly demonstrated in the paper. It would be more convincing if the authors can provide analysis results of transcriptome data. Also, it would be interesting to explore whether the transcription is affected once the grid-pattern disappears, or vice versa.*

We agree that the relationship between transcription and the interaction patterns is not investigated in detail in our manuscript, and thus to better represent the scope of our study, we have removed this emphasis from the abstract. To generally characterize the grid-pattern, we added an analysis where we called Hi-C peaks, and look at the enrichment of Rec8 sites around Hi-C peak anchors. This is greatly enriched, confirming our previous observations. We included this analysis in a revised **Supplementary Fig. 4a**, and refer to it in the text on lines 110-111.

To provide an initial characterization of the correspondence between the grid-pattern and transcription, we also added an analysis of TSS orientation around Hi-C peak anchors in **a new main figure Fig. 6d,e**. This analysis shows an enrichment of convergently-oriented genes around both meiotic and mitotic Hi-C anchors, the latter to a lesser extent perhaps consistent with the ability to call loop anchor positions in mitotic datasets being less efficient. We additionally found that the correspondence with the expression levels of meiotic genes, as assayed previously (Brar et al., *Science*, 2012), was interesting, yet inconclusive (see attached, below), thus we have chosen to not include this within the revised manuscript.

c) Frequency of TSSs by orientation around meiotic Hi-C peak anchors, in 500 bp bins smoothed with a sliding window of 3 bins to emphasize the enrichment patterns. d) As in c but stratified by expression levels in Leptotene/Zygotene (Brar et al., *Science*, 2012). Lower panels are smoothed with a 3-bin window to emphasise the enrichment patterns.

5. *Recently, several papers explored meiotic chromatin organization with Hi-C in mouse and monkey (Wang et al., Mol. Cell; Alavattam et al., NSMB; Patel et al., NSMB). It would be helpful if the authors can comment on the commonalities and differences between species.*

We have added a discussion of the current literature that has been published since the initial submission of our manuscript. In particular, we added a paragraph to the introduction (lines 54-64) and also discuss our findings in the context of the papers mentioned above in the results and discussion sections (lines 114, 230-238, 290-295)

6. *The title may be somewhat general, especially considering the differences in meiotic chromatin assembly among different species.*

In light of recent studies we have clarified our title to reflect the system we characterise:

“Principles of meiotic chromosome assembly in *Saccharomyces cerevisiae*”

Minor comments:

7. *Extended data Fig. 2, why does the cen-cen show higher trans interaction in *ndt80* mutant yeast compared with WT at 0 and 4 hour?*

This difference is most probably due to a small difference in cell culture between these particular wild type and *ndt80*Δ samples. Ndt80 does not have a role in vegetative cell growth (Xu et al., *MCB*,

1995) and is therefore unlikely to affect centromere interactions through a biological mechanism. We have added a comment for explanation to the legend of **Supplementary Fig. 2**.

8. *Fig. 2g, should the lines be blue instead of red?*

We thank the reviewer for the opportunity to clarify this figure. Briefly, in the **Fig. 2g** analysis we extract snippets of the Hi-C contact maps centered on Rec8 sites, and compute their average observed/expected ratio. This analysis shows that contact frequencies are enriched (red) between Rec8 sites and upstream/downstream regions but not across Rec8 sites, which demonstrates an insulation at these sites. To address this point, we have restructured the figure legend, placing interpretations for sub-panels when the panel is introduced instead of a collective summary at the end, and hope that this provides more clarity.

9. *Fig. 4a-b, can the author also provide a statistical result for the zip1 and hop1 deficient cells, similar as Fig. 1d?*

We have added this as **Supplementary Fig. 2c** showing the cis/total interactions as in **Fig. 1d**, for all analyzed mutants.

10. *Some figure orders need to be fixed. Fig. S4 appeared ahead of Fig. S3. Fig. S4b-c appeared prior to S4a.*

We thank the reviewer for pointing this out. We have made the corresponding changes.

Reviewer 2:

1. *Regarding novelty, there are several interesting results here but it is not clear enough that some of this has been done already: a previous paper by Muller et al Mol Syst Biol 2018 has measured yeast meiosis Hi-C (including the npt80 mutant which the authors here used), using a system of synthetic designed chromosomes which might even be considered superior for some of these questions. This previous work also showed the anchors at rec8 sites. The authors do cite this paper on the matter of rec8 sites. I suggest better clarifying in the text what has been already shown versus the new results.*

We thank the reviewer for the opportunity to clarify the new results in our manuscript. Muller et al. developed a promising approach to study the behavior of homologous chromosomes, by engineering a synthetic genomic region on chrIV to disambiguate reads coming from either chromosome, and performing Hi-C. Their observation that Rec8 sites show insulation is confirmed, and emphasised by our detection of a much stronger insulation effect reported in **Fig. 2g** in our manuscript. We extend this observation by detecting punctate grid-like interactions that correlate with Rec8 sites, and which are more prominent than the locus-specific folding observed previously. Importantly, we demonstrate that Rec8 is absolutely required for the emergence of these interactions. In addition to reporting observed in Hi-C maps, we use polymer simulations to quantitatively test mechanisms by which cohesin could establish such prominent interactions. These analyses clearly distinguish our manuscript from the findings reported by Muller.

To address these points, we refer to Muller et al. in the introduction and results sections, and added higlass browser links (below) to allow readers the opportunity to compare, side-by-side, the two datasets, which we hope clarifies the advances made within our study:

http://higlass.pollard.gladstone.org/app/?config=NKoclcPJRTuah4ZrQPPm_Q

<http://higlass.pollard.gladstone.org/app/?config=PrzfKuc2TwWu4rlfYaRH4w>

2. *It is was unclear to me how goodness of fit is measured, and what the final quality of fit to the data is, for example in Figure 3d. It would also be helpful to provide an additional metric for the fit*

between these maps, e.g. Pearson correlation. Finally, it is important to test these metrics after normalizing the maps for genomic distance.

We thank the reviewer for the opportunity to clarify our methodology. For a polymer model there are two important features of Hi-C data to recapitulate: both the overall dependence on genomic distance captured by the $P(s)$ curve, and, if possible, the locus-specific structure. Comparing each of $P^{Rec8-Rec8}(s)$, $P^{Rec8-non}(s)$, and $P^{non-non}(s)$ to their experimental counterparts captures both of these quantities and naturally normalizes for genomic distance. We now clarify in the **Fig. 3** legend (lines 400-403). Additionally, in the **Methods** we added an equation describing how to calculate this goodness-of-fit (Lines 743-746).

3. *Following on the previous point, was there some kind of cross validation here? It is difficult for the reader to evaluate whether there is any overfitting here. To ensure there is no overfitting, it would be good to estimate parameters on one region and apply the model to a different region using these parameters and the rec8 sites for the new region.*

We thank the reviewer for this suggestion. While exploring the full parameter space with locus-specific genome-wide simulations was overly computationally expensive, it is indeed feasible to generate genome-wide simulations for a single parameter set. Thus, to address this point, and better understand the properties of our best-fitting model, we performed new genome-wide simulations. Briefly, we ran simulations for best-fitting loop extrusion parameters found previously for chromosome 13 for the full genome, representing each chromosome as a polymer fiber and taking positions of barriers for Rec8 sites across the genome. This analysis shows that for most chromosomes, the goodness-of-fit is as good as found previously for chromosome 13. We describe this analysis in a new **Supplementary Fig. 6**.

4. *Regarding the parameters of the model, if I understand correctly the authors re-estimate the parameters for the rec8 mutant. Doesn't it make more sense to keep the loop extrusion parameters the same?*

To find best-fitting parameters for a given experimental dataset, we calculate the goodness-of-fit between that experimental dataset and simulated Hi-C data derived from loop extrusion dynamics with a wide range of parameter combinations. To clarify, we now note in the **Supplementary Fig. 5e** legend (Lines 542-543) that:

"parameters in best agreement with experimental ndt80Δ Hi-C data fit the experimental rec8Δ Hi-C data poorly (Average log(fold deviation)>2)."

We also clarify our approach in the main text (Lines 152-154) as: *"To find loop extrusion dynamics in agreement with experimental data, we computed the goodness-of-fit between experimental ndt80Δ Hi-C maps and simulated Hi-C maps generated for a wide range of parameter combinations (Fig. 3, Methods)."*

5. *It would be useful if the authors could provide a quantitative estimate of how synchronized the cells are, e.g. what fraction is in pachytene.*

The purity of our cell culture has been determined by monitoring completion of DNA replication (FACS) and using *ndt80Δ*, which prevents exit from pachytene. Following these measurements we aimed to have at least 90% of the population with duplicated DNA (which usually happens within 2-4 h, see **Fig. 1b,f**) and we incubated the cultures for 8 h total in sporulation media to allow the cells to reach pachytene (which is reached by the majority of cells by 4 h in wild type cells, see **Fig. 1a,e**). We also now provide FACS data for *rec8Δ*, *hop1Δ*, and *zip1Δ* samples (**Supplementary Fig. 2d**).

6. *The authors suggest the rec8-anchored patterns are closer to focal interactions than to TADs. Looking at the figures this is not obvious, and it also seems somewhat different from the data of Muller et al. Is there a more quantitative way of making this claim, for example by 2d peak calling? At minimum I suggest showing a better figure of this. In any case, the distinction between the two cases*

does not seem that clear given that the underlying mechanism suggested here (loop extrusion) is the same mechanism used to explain TADs.

The reviewer makes a good point to clarify what we meant, as the term TAD has been used to describe a range of square-like features in the literature. To address this, we extended panels **Fig. 3d-e** showing a set of parameters that gives rise to patterns that visually appear more like TADs, in that they are square-like rather than punctate, yet quantitatively agree poorly with our meiotic *ndt80Δ* data (position **iv**). We refer to this in the main text (lines 186-192). We also emphasize how a single underlying mechanism can give rise to different patterns in the Hi-C map depending on the parameters describing cohesin metabolism (lines 278-285).

7. *In Figure 2b, it seems like domain structure remains that is aligned with rec8 sites is present in the rec8 ndt80 mutant. There seems to be no mention or explanation of this in the text.*

Thank you for this remark. Indeed there seems to be a remaining structure in *rec8Δ* mutant cells. However, quantitative data analysis indicate that this is not a global trend displayed at Rec8 sites (see **Figure 2f,g**). We have added a comment regarding this observation to the figure legend (lines 388-389).

Reviewer 3:

1) *Line 62: Although telomere clustering is not extensive, bouquet formation is indeed conserved in budding yeast. Therefore, the description should be changed; e.g. telomere clustering was not detected.*

We have changed this sentence accordingly (lines 73-76):

*“Subtelomeric clustering also decreases during meiotic prophase (**Fig. 1a,d, Supplementary Fig. 1a, Supplementary Fig. 3**). Our wild-type timecourse displayed no evidence of a telomere bouquet, likely due to its transience, which has been measured by microscopy (Hayashi et al., 1998, #80514).”*

2) *Line 79-82: It is not clear whether only distinct behavior of subtelomeric and subcentromeric regions explains the different behavior of long and short arms. Namely, the short arm might be more influenced by them but not other unknown mechanism. Is this true?*

We agree we have yet to disentangle these possibilities and have clarified the text accordingly in response to this point (lines 94-99):

*“These features may arise from the distinct behavior of subtelomeric and subcentromeric regions (**Fig. 1h, Supplementary Fig. 1g**). Alternatively, or in addition, distinct P(s) for chromosomes with different length arms (**Supplementary Fig. 1h**) may be due to the centromere insulating the process that leads to differences between arms. In agreement with this, compaction is interrupted at centromeres in Hi-C maps (**Fig. 1a, Supplementary Fig. 2b**).”*

3) *Line 83-86: Regarding to the above point, the last two sentences of this section might be not significant.*

We have clarified the text accordingly in response to this point (see above).

4) *Line 174-186, 212-219: A recent study shows that Rec8 cohesin complex directly bind Rec10 (Red1 homolog) in early prophase in fission yeast meiosis (Dev Cell 32, 220-230, 2015). Also in budding yeast, Red1 localizes at chromosome axis prior to Hop1 or other axis components (JCB 136, 957-967, 1997). Therefore, Rec8 cohesin complexes sitting at the barrier sites may be stabilized by Red10 (even without Hop1 and Zip1). This could be experimentally examined or mentioned as a possibility.*

This is an interesting proposition. Indeed, we are continuing this project by investigating more closely the effects on loop positioning by several axis components (including Red1). We feel however that this work will be a continuation or separate part rather than an additional part to this manuscript. As

suggested by the reviewer, we have now added the potential distinct role for Red1 to the discussion (Lines 264-267).

Reviewer 4:

Major points

1) *The structure of the paper is not conventional for Nature Communications. In particular, the Introduction section is merged with the Results. This choice makes it difficult to easily distinguish the novelties of this work from the results in the literature. It would add to the clarity of the manuscript if the authors properly divide these two sections.*

We have followed this suggestion and clearly divided the sections of **Introduction** and **Results**. Additionally, we have added subheadings to the results sections to aid readability.

2) *The division of Introduction and Results sections would help the authors to aptly comment on existing literature that is currently completely missing or poorly cited. Specifically, the authors should comment on the three recent publications (Alavatam et al., 2019, Nature Structural & Molecular Biology 26, 175–184; Patel et al., Nature Structural & Molecular Biology 26, 164–174 (2019), Wang et al., 2019, Molecular Cell 73, 547–561) that report on Hi-C experiments of meiotic chromosomes in three different species human, mouse and rhesus monkey respectively. Properly commenting on these published works would be beneficial for the community to put this work in the correct context and for the authors to fully highlight the novelties of their work.*

We thank the reviewer for the opportunity to cite and discuss the many important papers that came out after our initial submission. In our revision, we added an overview of the findings of relevant recently published literature to the introduction (lines 54-64) and have specifically referred to the individual papers in results and discussion where relevant.

3) *The experiments and the simulations are quite sound to me. However, some details of the simulation protocol are currently missing. Addressing the following points would add on the reproducibility of their computational work:*

- *At page 20 line 541 the sentence “We simulated the chromatin fiber with excluded volume interactions and without topological constraints” should be clarified in particular about the absence of topological constraints. Does it mean that the chains are phantom? In this case how this fits with the applied excluded volume interaction? Are the bonds crossable? Or does it mean that the chains are unknotted in the initial state? In this case, how do the authors check the initial unknotted state and its persistence along the entire simulation?*

As in previous work from us (Schalbetter et al., NCB, 2017) and others (e.g. Zhang & Wolynes, PNAS, 2015), we used a soft-core excluded volume potential which allows occasional chain passing. We now clarify this in the **Methods** (Lines 688-691).

- *What is the starting conformation of the simulated polymer chains? How does it affect the final result? I find this an important aspect, in particular, if the chains are not phantom.*

As for previous loop extrusion simulations from us (e.g. Fudenberg et al., Cell Reports, 2016) and ‘effective equilibrium’ simulations from others (e.g. Zhang & Wolynes, PNAS, 2015), we sample conformations at steady-state, averaged over many starting conformations and sufficient time such that Hi-C maps for each parameter set do not depend on the exact starting conformation. We now clarify this in the **Methods** by noting we collect conformations “at steady state” (line 732).

- *What is the size of the nucleus and, as a consequence, the density of the polymer solution? Is it chosen following experimental evidence?*

We now clarify in the **Methods** (lines 687-688) that polymers are: “confined to a 1 micron radius nucleus”, as in previous simulations of yeast genomes (Tjong et al., *Genome Res.*, 2012; Wong et al., *Current Biology*, 2012).

- *What contact radius has been used to compute the contact maps of the simulated systems?*

We now clarify in the **Methods** (lines 734-735) as: “Contacts were recorded between any two monomers in a given conformation separated by less than 60nm.”

- *The visual similarity of the simulated (best-fitting parameters) vs experimentally measured contact maps is usually striking in the figures of the manuscript. However, when the authors compare the models with the experimental results, they rather prefer to examine the contact probability over the genomic distance ($P(s)$) that is somehow a coarser quantity. Although the comparison is still acceptable for validation, the authors should comment in the manuscript on this particular choice.*

We note that we compare three classes of $P(s)$, that allow for quantifying the agreement with overall polymer compaction as well as locus-specific folding: $P^{Rec8-Rec8}(s)$, $P^{Rec8-non}(s)$, and $P^{non-non}(s)$, and note that we have expanded our **Methods** section (lines 743-746), as mentioned in reply to Reviewer 2 (point 2).

Some minor points need to be clarified or corrected. In particular:

- *In page 2, line 58, Supplementary Fig. 2 should read Supplementary Fig. 2.*

Thank you, we have incorporated this correction.

- *The panels in Figure 3c and 4e both represent the Goodness-of-fit, but their colour scales have different labels.*

We have now changed the labels throughout the manuscript to be consistent.

- *The panels in Figure 2g and 4f are not referenced in the main text, and the Extended Data Figure 2c that is referenced in page 6 line 177 is missing.*

We have added a reference to **Fig 1g** which was missing (line 96). We have revised the structure of **Fig. 4** (now **Fig. 5**) and moved panel **4f** to **Fig. 6a**. We have corrected the reference to ‘**Supplementary Fig. 2b,c**’ to state ‘**4b,c**’ (line 215).

REVIEWERS' COMMENTS:

Reviewer #1 (Remarks to the Author):

The authors have well addressed my previous concerns. I do not have further comments and I congratulate the authors for this elegant study.

Reviewer #2 (Remarks to the Author):

The authors have done a good job clarifying the paper and addressing my concerns.

Reviewer #4 (Remarks to the Author):

The authors have answered all of the raised points satisfactorily, and have written a very interesting, informative paper. I recommend it for publication in Nature Communications.